# Coreset-Driven Re-Labeling: Tackling Noisy Annotations with Noise-Free Gradients

**Saumyaranjan Mohanty**                                                          *ai23resch04001@iith.ac.in*
*Department of Artificial Intelligence*
*Indian Institute of Technology Hyderabad*

**Konda Reddy Mopuri**                                                          *krmopuri@ai.iith.ac.in*
*Department of Artificial Intelligence*
*Indian Institute of Technology Hyderabad*

**Reviewed on OpenReview:** *https: // openreview. net/ forum? id= Tk78vb2Qd7*

## Abstract

Large-scale datasets invariably contain annotation noise. Re-labeling methods have been developed to handle annotation noise in large-scale datasets. Though various methodologies to alleviate annotation noise have been developed, these are particularly time-consuming and computationally intensive. The requirement of high computational power and longer time duration can be drastically reduced by selecting a representative coreset. In this work, we adapt a noise-free gradient-based coreset selection method towards re-labeling applications for noisy datasets with erroneous labels. We introduce 'confidence score' to the coreset selection method to cater for the presence of noisy labels. Through extensive evaluation over CIFAR-100N, Web Vision, and ImageNet-1K Datasets, we demonstrate that our method outperforms the SOTA coreset selection for re-labeling methods (DivideMix and SOP+). We have provided the codebase at URL.

## 1 Introduction

The rise in the size and complexity of modern datasets and deep learning models has resulted in the use of extensive computational resources. Modern deep-learning tasks depend upon very large datasets to achieve State-Of-The-Art (SOTA) performance (Pooladzandi et al., 2022; Yang et al., 2023). Supervised learning tasks such as image classification require labels for each image in the training and test sets. Dataset annotation/labeling is usually done through human annotators. The human factor invariably leads to noisy labels in the datasets. In turn, these noisy labels pose a significant challenge to training deep learning models and severely degrade the generalization performance of deep neural networks (Song et al., 2022). Some of the popular approaches towards dealing with noisy labels involve the development of robust architectures (Xiao et al., 2015), unbiased estimators and weighted loss functions (Liu & Tao, 2016; Natarajan et al., 2013), robust regularizers (Shorten & Khoshgoftaar, 2019), loss correction (Liu & Guo, 2020; Patrini et al., 2017), sample selection-aided (Han et al., 2018b; Jiang et al., 2018; Yu et al., 2019), etc.

*Re-labeling* (Song et al., 2019) is a family of robust training methods to identify wrong labels and correct these labels during training time. Consistency training methods (Xie et al., 2020a) apply regularization to model prediction to ensure that the model is invariant to sample noise. These self-consistency regularisations have been shown to achieve SOTA performance on the re-labeling tasks. Li et al. (2020b) introduced DivideMix, a framework for learning with noisy labels that leverages semi-supervised learning techniques. Sparse Over-Parameterization (Liu et al., 2022a) proposes a principled approach for robust training of deep learning models for noisy training labels. This method assumes label and model noise are sparse and trains to separate noise from data. Due to various factors such as augmentations and multiple backbones, these methods require high computational time.

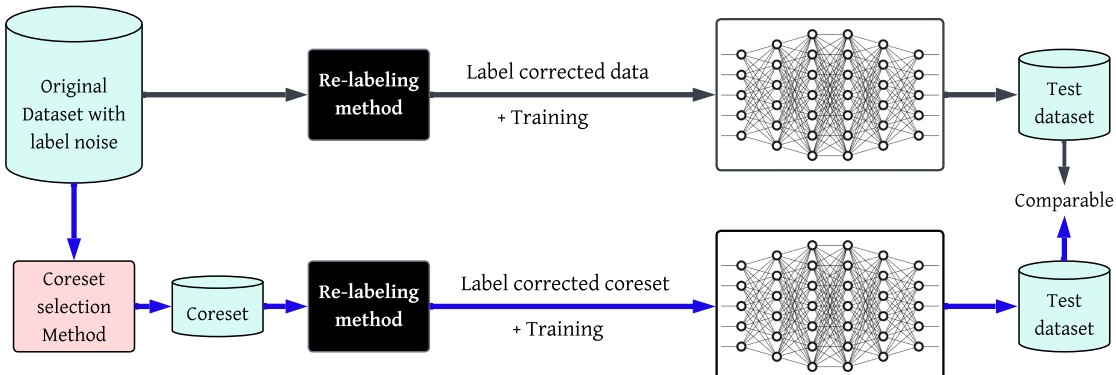

Figure 1: Block diagram of the entire process. The re-labeling method is used as a black-box. We have empirically shown that selection of a much smaller coreset through our proposed method improves training speed while not significantly impacting the generalization performance of the output of the re-labeling method over the coreset.

To reduce the computational burden, Park et al. (2023) have introduced utilization of coreset with re-labeling for training a deep neural network over a dataset with noisy labels. They have shown that existing data-pruning/coreset selection algorithms don't work well with re-labeling task because they don't take re-labeling into account. They have proposed a method called *Prune4Rel*, which selects a subset that maximizes the re-labeling capability.

Coresets aim to address the extensive compute and storage requirements for training complex deep learning models (CNNs, transformers, foundation models, etc.) on large-scale datasets. Training these complex models on large datasets requires very high computational power and is very time-consuming as well (Dong et al., 2021). Hyperparameter tuning of these complex deep-learning models is also computationally intensive (Zoph et al., 2018) and requires a lot of time to learn a set of hyperparameters that optimize the model for the task at hand. For instance, the storage requirement for large-scale visual benchmark datasets like ImageNet-22k (Deng et al., 2009) is in the order of TBs (Xia et al., 2023). Furthermore, the advent of complex models and very large datasets has led to an exponential rise in carbon footprint (tec; Killamsetty et al., 2021c).

Coreset Selection aims to mitigate these issues by finding the most representative subset of the original larger dataset. In particular, coreset selection methods attempt to approximate the learning characteristics of the complete dataset (e.g., gradient information) (Feldman, 2020). A representative coreset with the cardinality of a fraction of the entire dataset would drastically reduce training duration and computational requirements for end-to-end training while delivering the desired generalization performance. Sorscher et al. (2022) have shown that the discovery of good data-pruning metrics may provide a viable path forward to substantially improved neural scaling laws, thereby reducing the resource cost of modern deep learning.

To study the effectiveness of the coreset approach for datasets with noisy labels, we begin with analysis of CIFAR-100N (Wei et al., 2022), a dataset specifically created to underscore the presence of noisy labels due to human annotation errors. Figure 2 depicts a bar plot of the number of mislabeled samples in class 'Apple' and the original class they belong to. It can be seen that, of all the mislabeled samples, 25% belong to the class 'pear' and 21% belong to the class 'sweet pepper', two of the visually similar classes.

This similarity has motivated us to utilize an intuitive coreset selection method based on 'Noise-free Loss Gradients' introduced by Mohanty et al. (2025) and apply this to the task of re-labeling on a noisy dataset. This approach uses the similarity of loss gradients to compose the coreset. We have adapted their approach by introducing confidence score-based weights to calculate gradient similarities. Through extensive experimentation over various complex, noisy datasets, we have shown that the coreset selected through our method outperforms all other coreset selection methods on the re-labeling model training task (including *Prune4Rel*,

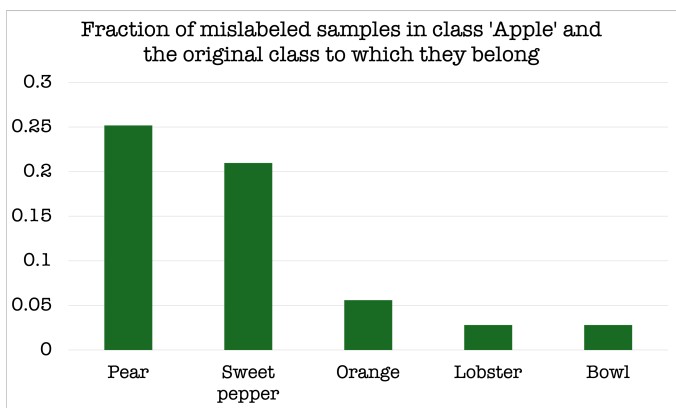

Figure 2: Fraction of mislabeled samples in class 'Apple' and the original class they belong to. The top two mislabeled classes are 'Pear' and 'Sweet Pepper', which are visually similar and can be mistaken by a human annotator.

which is specifically designed for this task). The re-labeling method is treated as a black-box, with comparison being carried out between the performance of a model trained on label-corrected data from the original dataset and a model trained on label-corrected coreset from a coreset composed from the original dataset through our method. Figure 1 shows the block diagram of the process.

In summary, the major contributions of our work are as follows.

- Repurpose an existing gradient similarity-based coreset selection method to achieve generalization in the presence of noise (via re-labeling).

- Leverage model confidence as a proxy for reliability of the samples, enhancing the robustness of the selected coreset.

- Thoroughly evaluate the proposed method over multiple noisy datasets for the re-labeling task on image classification.

## 2 Related Works

In this section, we briefly discuss the prominent works in the field of coreset selection and re-labeling.

### 2.1 Re-labeling

The massive labeled datasets are the backbone of the impressive performance of deep neural networks. But using crowd-sourcing marketplaces such as Amazon Mechanical Turk (Turk) to reduce labeling cost has resulted in unreliable labels (Scott et al., 2013). These noisy labels lead to poor generalizability on the test dataset (Zhang et al., 2017). Song et al. (2020) has presented a detailed survey of deep learning approaches with noisy labels. The ratio of corrupted labels in real-world datasets ranges from 8.0% to 38.5%.

Goldberger *et al.* (Goldberger & Ben-Reuven, 2017) introduced a noise adaptation layer into the deep neural network, an additional softmax layer that explicitly models the correlation between correct labels and noisy ones. A human-assisted approach called "Masking" was introduced by Han et al. (2018a) for estimating the noise transition matrix. This method incorporates a structure-aware model, making the transition matrix estimation more effective. A quality-embedding model was proposed by Yao et al. (2019) to improve the deep learning model's ability to learn from datasets with unreliable annotations.

While the methods mentioned in the previous paragraph explored architecture modifications, Ghosh et al. (2017) proposed a robust loss function to mitigate the effects of label noise in deep learning. Hendrycks et al. (2018) proposed that a small subset of trusted training data would be able to achieve substantial robustness

performance gain on noisy labels. A normalized loss function was introduced by Ma et al. (2020) to make the loss function robust to noisy labels. Termed as "Active Passive Loss", it combines two robust loss functions that mutually boost each other.

Zhou et al. (2021) introduced a dynamic curriculum learning approach, which dynamically transitions from learning clean labels to self-supervised learning with pseudo labels. The approach improves label selection and generalisation by tracking training dynamics across multiple steps and augmentations. These 'Re-labeling' approaches try to recover correct labels from noisy labels through heuristic rule (Song et al., 2019). Two recent SOTA works on re-labeling are DivideMix (Li et al., 2020a) and SOP+ (Liu et al., 2022b). DivideMix simultaneously trains two divergent neural networks using dataset division from the other network. SOP+ exploits the sparsity of the label noise by modelling the label noise via another sparse over-parametrization term. All these approaches utilize self-consistency loss to leverage strong augmentation to implicitly correct noisy labels. Due to the introduction of additional architectures and the requirement of strong data augmentations, re-labeling models require more computation power and execution time (Chen et al., 2019).

## 2.2 Coreset selection

In this sub-section, we briefly review some influential works on selecting representative subsets to achieve better generalization performance. Various works have proposed different approaches to selecting a coreset.

Uncertainty-based methods propose that samples with lower confidence scores on a model are more impactful for model generalization than samples with higher confidence. Selection via Proxy method (Coleman et al., 2020) uses a smaller proxy model instead of full-scale target models. These smaller and faster models provide useful signals for coreset selection based upon uncertainty and representativeness. Other metrics used to calculate sample uncertainty are least confidence (Shen et al., 2018) and entropy (Settles, 2012).

Loss/Error-based methods select coresets based on each sample's contribution towards the loss function during model training. Catastrophic forgetting was used by Toneva et al. (2019a) to select a coreset composed of forgettable samples. If the model correctly classifies a sample through multiple epochs, it can be removed from the dataset with minimal performance drop. Two metrics called GRAND and EL2N (Paul et al., 2023) measure the average contribution from each sample. This helps in pruning significant fractions of training data without sacrificing test accuracy.

Decision boundary-based methods choose the data points closest to the decision boundary as the coreset. Adversarial Deepfool (Ducoffe & Precioso, 2018) introduces closeness to the decision boundary by introducing perturbations to the samples that result in a change in label predictions. Contrastive Active Learning (Margatina et al., 2021) measures closeness through divergence of predictive likelihood.

Geometry-based methods select coresets by removing data points clustered in the feature space. Herding (Chen et al., 2010) selects a coreset by greedily adding samples to the coreset that minimize the distance between the coreset center and the dataset center in the feature space. K-Center Greedy approximation (Sener & Savarese, 2018) attempts to solve the minimax facility location problem to select coresets from a large dataset such that the maximum distance between points in the non-coreset and their closest point in the coreset is minimized.

Gradient Matching based methods such as CRAIG (Mirzasoleiman et al., 2020) and GRADMATCH (Killamsetty et al., 2021a) utilize gradients produced by the full training dataset and select a coreset whose weighted gradients would result in the minimal difference. RETRIEVE (Killamsetty et al., 2021c) and GLISTER (Killamsetty et al., 2021b) pose the coreset selection problem as a bi-level optimization problem, which treats the selection of a subset as the outer objective and optimization of model parameters as the inner objective. An intuitive coreset selection method termed 'Noise-free Loss Gradients' based on the similarity of loss gradients was proposed by Mohanty et al. (2025), which composes a coreset of samples that have the maximum number of neighbours with higher cosine similarity among their gradients.

Submodular functions (Iyer & Bilmes, 2013) have been utilized to measure diversity and information and have been incorporated for coreset selection by various methods such as graph cut and facility location. Moderate method (Xia et al., 2023) calculates the distance between the hidden representation of samples

and the representational class centres. Based on these Euclidean distances, they rank the data points in ascending order and select the data points closest to the distance median as a coreset. An influence function-based iterative method was proposed by Yang et al. (2023), which calculates the influence of each sample on the model's parameter training and composes the coreset with samples that have the most influence.

All these works have not considered the label noise inherent in larger datasets. Prune4Rel (Park et al., 2023) proposes a data pruning algorithm based on the maximization of total neighbourhood confidence of all training examples to ensure the method selects the samples that maximize the re-labeling accuracy, thereby resulting in better generalization performance.

## 3 Methodology

### 3.1 Detailed Methodology

This section provides a detailed description of the proposed approach. We adapt the concept of 'Noise-free Loss Gradients' (Mohanty et al., 2025) and introduce an additional per-sample confidence score on top of the existing coreset selection method to capture the representation ability of the samples of the dataset. Confidence score refers to the probability assigned by the model to each sample of belonging to a given class.

As we have brought out in Section 1, major annotation mistakes happen within visually similar classes. Table 1 presents a few classes from the CIFAR-100N dataset, with the two dominant classes to which most mislabeled samples belong. This analysis shows that most mislabeling occurs because these classes are visually similar to the incorrect class. A model trained on a noisy dataset tends to assign higher confidence to mislabeled images that are visually similar to the incorrect class than to those that are not (Figure 3). Therefore, if we can select a representative coreset that includes samples belonging to these dominant mislabeled classes, they would have a higher chance of getting corrected by the re-labeling models.

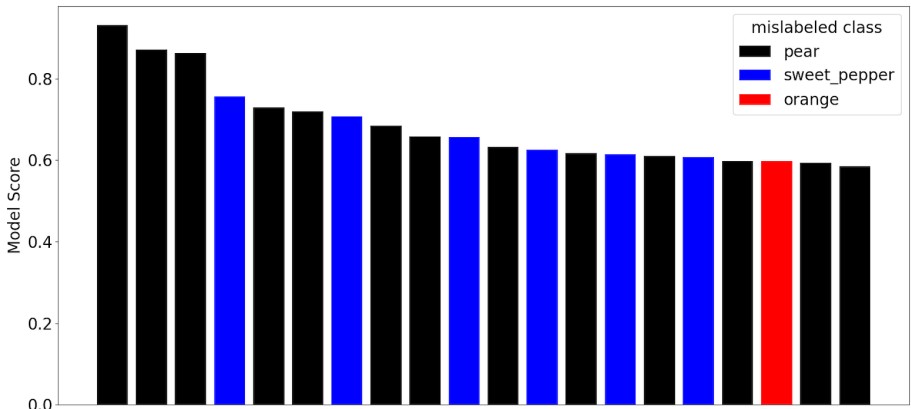

Figure 3: Confidence score assigned by a model to mislabeled samples belonging to class 'Apple'. The bar plot shows the top 20 mislabeled samples and their confidence score in descending order. The samples are color-coded by the real class they belong to. As we can notice, the model assigns higher confidence scores to mislabeled samples that are visually similar to 'apple'.

Table 2 provides the notation used throughout the paper.

$\mathbb{V}$ is the original training dataset with noisy labels. A re-labeling method tries to correct the noisy labels through minimizing the re-labeling loss function $L_{relabeling}$. As the re-labeling methods are quite computation intensive, our objective is to select a coreset $\mathbb{S}$ from $\mathbb{V}$, such that, the classification performance on the coreset after re-labeling closely matches the performance achieved by re-labeling the entire dataset.

Let $(x_i, y_i) \in \mathbb{V}$ be the original image and label pair and $(x_i, y_i^*)$ be the corresponding ground-truth label. Let $(\tilde{x}_i, \tilde{y}_i) \in \mathbb{S}$ be the coreset composed from $\mathbb{V}$ and $(\tilde{x}_i, \tilde{y}_i^*)$ be the corresponding ground-truth label. The

Table 1: Analysis of the top two classes to which most mislabeled examples belong. For samples belonging to the class 'Aquarium Fish', most of the mislabeled samples belong to the 'Trout' and 'Flatfish' classes, which are inherently similar and can be mistaken by a human annotator.

| Class name | Top two mislabeled classes |
|---|---|
| Apple | Pear(25%), Sweet pepper(21%) |
| Aquarium fish | Trout(35%), Flatfish(12%) |
| Bed | Couch(28%), Table(22%) |
| Bicycle | Motorcycle(30%), Lawn mower(22%) |
| Castle | House(35%), Skyscraper(14%) |

Table 2: Notation

| Symbol | Description |
|---|---|
| $\mathbb{V} = \{(x_i, y_i)\}$ | Large Training Dataset |
| $\mathbb{S} \subset \mathbb{V}$ | Coreset of $\mathbb{V}$ (target) |
| $\theta$ | Parameters of the classifier |
| $\mathcal{A}$ | Strong data augmentation |
| $\mathbb{L}_{ce}$ | Cross entropy loss function for classification |
| $\mathbb{L}_{relabelling}$ | Loss function for the re-labeling method |
| $\Phi$ | Threshold on the gradient similarity for neighbourhood identification |
| $\mathbb{V}_c$ | Training data belonging to class c |
| $g_{x_i}^\theta$ | Loss gradients computed for data sample $(x_i, y_i)$ at the last fully connected(classification) layer |
| $\|x\|$ | $l_2$ norm of $x$ |
| $\|x\|_1$ | $l_1$ norm of $x$ |
| $|\mathbb{V}|$ | Cardinality of set $\mathbb{V}$ |
| $\mathbb{1}$ | Indicator function |
| $< ., . >$ | Dot product operator |

re-labeling loss function with self-consistency regularization (Xie et al., 2020b) on coreset $\mathbb{S}$ is given by:

$$\mathbb{L}_{relabeling}(\mathbb{S}; \theta, \mathcal{A}) = \left[ \sum_{(\tilde{x}, \tilde{y}) \in \mathbb{S}} \mathbb{1}_{[C_\theta(\tilde{x}) \geq \delta]} \mathbb{L}_{ce}(\tilde{x}, \tilde{y}; \theta) \right] + \lambda \left[ \sum_{\tilde{x} \in \mathbb{S}} \mathbb{L}_{reg}(\tilde{x}; \theta, \mathcal{A}) \right] \quad (1)$$

where $\mathcal{A}$ is a strong data augmentation, $C_\theta(x)$ provides the prediction confidence score, and $\delta$ is the threshold for prediction to switch from a wrong label to a correct label upon administration of strong augmentation $\mathcal{A}$. Let $\mathbb{S}^*$ denotes the label corrected coreset. Our objective is to select a coreset $\mathbb{S}$, that will maximize performance of a classification model trained on label corrected coreset $\mathbb{S}^*$.

'Noise-free Loss Gradients' (Mohanty et al., 2025) measures representation ability of each sample $(x_i, y_i)$ in the dataset $\mathbb{V}$ as:

$$f(x_i, y_i) = E_\theta \left[ \sum_{(x_j, y_j) \in \mathbb{V}, \ j \neq i} \rho(x_i, y_i, x_j, y_j, \theta) \right] \quad (2)$$

Where $\rho$ is the normalized cosine similarity between gradients of two samples as given in Eq. 3.

$$\rho(x_i, y_i, x_j, y_j, \theta) = \frac{< g_{x_i}^\theta, g_{x_j}^\theta >}{\|g_{x_i}^\theta\| \|g_{x_j}^\theta\|} \quad (3)$$

A nearest neighbor search algorithm identifies the number of samples within a specified radius of each sample, that is, samples whose similarity exceeds a given threshold $\Phi$, and assigns it as its score. Samples are then ranked according to their aggregated scores across multiple checkpoints. For a given class $c$, the top-ranked samples $(x_k, y_k)$ are selected based on their scores.

$$(x_k, y_k) = \arg\max_{(x_i, y_i) \in \mathbb{V}_c} \sum_\theta \sum_{j \neq i} \mathbb{1}(\rho(x_i, y_i, x_j, y_j, \theta) > \Phi) \quad (4)$$

We introduce a weighted representation ability based on a sample confidence score. For the model parameter $\theta$ at a given checkpoint, confidence score of a sample $x_i$ belonging to class $c$ is given as:

$$C_\theta(x_i) = \theta(x_i)[c] \tag{5}$$

where $\theta(x_i)$ is the $k-$length output of the model parametrized by $\theta$ and $k$ is the number of classes in the dataset.

We modify Eq. 4 as:

$$(x_k, y_k) = \underset{(x_i, y_i) \in \mathbb{V}_c}{\arg\max} \sum_\theta \sum_{j \neq i} C_\theta(x_j) \mathbb{1}(\rho(x_i, y_i, x_j, y_j, \theta) > \Phi) \tag{6}$$

Using this weighted measure, we compute the suitability of every sample in $\mathbb{V}$ to become an element of the coreset $\mathbb{S}$ in a class-wise manner. Essentially, this translates to sorting the dataset samples in each class in decreasing order of this weighted measure and composing the coreset of a desired cardinality.

Algorithm 1 presents our approach more formally.

---

**Algorithm 1** Modified Noise-free Gradients for Re-labeling algorithm

---

**Require:** Train set: $\mathbb{V}$; Total epochs: $T$; number of classes: $C$; number of coreset images per class: $N$;
    Model checkpoint at all initial epochs up to T : $\theta$
**Ensure:** Coreset $\mathbb{S}$
1: **for** class $c$ in 1,..., $C$ **do**
2:     **for** $(x_i, y_i) \in \mathbb{V}_c$ **do**
3:         **for** epochs $t$ in 1,..., $T$ **do**
4:             compute $g_{x_i}^{\theta_t}, C_{\theta_t}(x_i)$
5:         **end for**
6:         $f(x_i) = \sum\limits_{\theta_t} \sum\limits_{(x_j, y_j) \in \mathbb{V}_c, j \neq i} C_{\theta_t}(x_i) \mathbb{1}(\rho(x_i, y_i, x_j, y_j, \theta_t) > \Phi)$
7:         Store $f(x_i)$
8:     **end for**
9: **end for**
10: $\mathbb{S} = \emptyset$
11: **for** class $c$ in 1,..., $C$ **do**
12:     $\mathbb{S} \leftarrow \mathbb{S} \cup \operatorname{argsort}_{(x_i, y_i) \in \mathbb{V}_c} f(x_i)[: N]$                            ▷ Descending order
13: **end for**

---

### 3.2 Comparison with Prune4Rel

**Mechanism**: Prune4Rel maximizes total neighborhood confidence to select coresets, focusing on local label consistency. Our method adapts the noise-free gradient framework for coreset selection, weighted by per-sample confidence scores.

**Confidence definition**: Prune4Rel uses neighborhood-based confidence, while we use softmax confidence (i.e., model confidence on each sample).

## 4 Experimentation

### 4.1 Experimental setup

**Applications**. We have evaluated our coreset selection methodology on re-labeling application for image classification over datasets with noisy annotations.

**Datasets**. We have evaluated the effectiveness of our method on three benchmark datasets.

CIFAR-100N (Wei et al., 2022) is the CIFAR-100 dataset with human-annotated real-world noisy labels collected from Amazon Mechanical Turk. This specialised dataset incorporates human-annotated real-world noisy labels. It consists of $50,000$ colour images of dimension $32 \times 32 \times 3$ from 100 different classes, each class having 500 images.

WebVision (Li et al., 2017) contains $2.4M$ images crawled from the Web using the $1,000$ concepts in ImageNet-1K (Deng et al., 2009). Similar to prior works (Chen et al., 2019; Park et al., 2023), we use the mini-WebVision version consisting of the first 50 classes of the Google image subset with approximately $66,000$ training images.

Following the approach in Park et al. (2023), we introduced 20% asymmetric noise to the ImageNet-1k (Deng et al., 2009) dataset. The noisy dataset is constructed by randomly selecting 20% of images from each class $c$ and flipping their labels to $c+1$. It is a subset of the larger dataset ImageNet, an image dataset organised according to the WordNet hierarchy. ImageNet-1K consists of 1000 classes, with $1,281,167$ training images and $50,000$ validation images.

**Re-labeling methods**. DivideMix (Li et al., 2020a) and SOP+ (Liu et al., 2022b) are used as the re-labeling methods.

**Coreset Methods**. We have carried out experimentation with various leading coreset selection methods, including k-CenterGreedy (Sener & Savarese, 2018), GraNd (Paul et al., 2021), Forgetting (Toneva et al., 2019b), SmallLoss (Jiang et al., 2018), Moderate (Xia et al., 2023) and Prune4Rel (Park et al., 2023). We have utilised the publicly available codebase of the Prune4Rel method[1] for conducting our experiments.

**Classifier**. Following prior works (Li et al., 2020a; Liu et al., 2022b; Park et al., 2023), we have used PreAct ResNet-18 (He et al., 2016b) architecture for CIFAR-100N, InceptionResNetV2 (Szegedy et al., 2016) architecture for WebVision and ResNet-50 (He et al., 2016a) architecture for ImageNet-1K dataset.

**Implementation**. Experiments for CIFAR-100N are carried out for five individual runs with different random seeds. Experiments for ImageNet-1K were carried out for two individual runs due to relatively higher computational requirements. We have used the same training strategy (optimiser, weight decay, batch size and other hyperparameters) as utilised by Prune4Rel.

**Hyper-parameter settings**. Table 3 tabulates various hyperparameter settings used in the experimentation for the ease of reproducibility.

Table 3: Hyper-parameter values used across multiple datasets.

| Settings | CIFAR-100N | WebVision | ILSVRC |
|---|---|---|---|
| Epochs | 300 | 100 | 50 |
| Optimizer | SGD | SGD | SGD |
| Momentum | 0.9 | 0.9 | 0.9 |
| Weight Decay | 0.0005 | 0.0005 | 0.0005 |
| Batch Size | 128 | 32 | 64 |
| Learning Rate | 0.02 | 0.02 | 0.02 |

**Codebase**. Codebase for implementing our method is provided at URL.

### 4.2 Results on CIFAR-100N

Table 4 tabulates results obtained on the CIFAR-100N dataset with the DivideMix Re-labeling method. The first column lists various coreset selection sizes as a fraction of the full dataset. Similarly, performance results obtained on CIFAR-100N dataset with SOP+ Re-labeling method is given in Table 5.

---

[1] https://github.com/kaist-dmlab/Prune4Rel

Table 4: Performance comparison of various coreset selection methods on CIFAR-100N dataset with DivideMix Re-labeling method at various selection fractions of the original dataset. The best results are shown in bold. For selection sizes ranging from 5% to 50%, our proposed method is able to beat Prune4Rel by average 4.22%.

|  | k-CenterGreedy | GraNd | Forgetting | SmallLoss | Uniform | Prune4Rel | Moderate | Ours |
|---|---|---|---|---|---|---|---|---|
| 0.05 | 13.88 ± 1.24 | 6.95 ± 0.86 | 11.18 ± 1.57 | 14.18 ± 1.13 | 13.56 ± 1.21 | 18.89 ± 0.23 | 10.05 ± 2.81 | **22.08 ± 0.17** |
| 0.10 | 22.69 ± 3.53 | 10.19 ± 0.32 | 19.64 ± 0.37 | 22.53 ± 1.43 | 21.89 ± 0.56 | 21.31 ± 2.01 | 16.85 ± 1.91 | **28.87 ± 0.96** |
| 0.20 | 38.01 ± 1.01 | 15.51 ± 1.21 | 26.41 ± 1.31 | 33.31± 3.22 | 30.52 ± 1.01 | 39.41 ± 0.81 | 34.21 ± 1.42 | **43.57 ± 1.25** |
| 0.30 | 44.18 ± 2.02 | 22.79 ± 1.78 | 48.44 ± 0.82 | 42.72 ± 3.62 | 43.65 ± 1.32 | 47.35 ± 1.36 | 42.65 ± 0.99 | **51.97 ± 1.62** |
| 0.40 | 50.07 ± 1.81 | 26.01 ± 1.92 | 54.33 ± 0.81 | 47.45 ± 1.13 | 55.31 ± 0.52 | 56.38 ± 0.52 | 54.57 ± 1.36 | **58.72 ± 0.34** |
| 0.50 | 54.15 ± 1.47 | 44.06 ± 0.84 | 57.34 ± 0.13 | 53.13 ± 1.49 | 53.97 ± 0.43 | 55.63 ± 1.08 | 51.77 ± 0.15 | **59.07 ± 0.23** |
| 0.60 | 59.74 ± 1.33 | 44.74 ± 1.54 | **63.17 ± 1.30** | 59.41 ± 0.71 | 57.58 ± 1.92 | 63.59 ± 0.34 | 56.18 ± 0.55 | 60.97 ± 0.19 |

Table 5: Performance comparison of various coreset selection methods on CIFAR-100N dataset with SOP+ Re-labeling method at various selection fractions of the original dataset. The best results are shown in bold.

|  | k-CenterGreedy | GraNd | Forgetting | SmallLoss | Uniform | Prune4Rel | Moderate | Ours |
|---|---|---|---|---|---|---|---|---|
| 0.05 | 21.29 ± 0.02 | 6.27 ± 0.34 | 22.73 ± 0.78 | 26.80 ± 0.58 | 24.99 ± 0.54 | 32.61 ± 0.29 | 22.38 ± 0.35 | **40.48 ± 0.02** |
| 0.10 | 33.11 ± 0.11 | 9.01 ± 0.55 | 27.73 ± 0.27 | 35.07 ± 0.51 | 32.87 ± 1.06 | 44.55 ± 0.06 | 27.69 ± 0.67 | **46.82 ± 0.34** |
| 0.20 | 46.62 ± 0.31 | 13.84 ± 0.33 | 43.55 ± 0.08 | 48.41 ± 0.36 | 46.98 ± 0.37 | 52.41 ± 0.44 | 44.51 ± 0.28 | **53.92 ± 0.24** |
| 0.30 | 52.31 ± 0.22 | 19.37 ± 0.17 | 51.21 ± 0.36 | 55.39 ± 0.39 | 52.72 ± 0.38 | 56.27 ± 0.07 | 50.82 ± 0.23 | **58.04 ± 0.25** |
| 0.40 | 56.73 ± 0.14 | 27.47 ± 0.37 | 56.54 ± 0.11 | 59.72 ± 0.25 | 56.45 ± 0.13 | 59.33 ± 0.11 | 54.61 ± 0.18 | **60.55 ± 0.17** |
| 0.50 | 59.48 ± 0.14 | 36.32 ± 0.31 | 60.68 ± 0.21 | 62.69 ± 0.14 | 59.03 ± 0.18 | 61.99 ± 0.19 | 58.07 ± 0.16 | **62.76 ± 0.17** |
| 0.60 | 61.97 ± 0.16 | 45.37 ± 0.07 | 63.38 ± 0.07 | 64.28 ± 0.25 | 61.11 ± 0.24 | 63.48 ± 0.51 | 60.21 ± 0.41 | **64.38 ± 0.05** |

## 4.3 Results on WebVision

Table 6 tabulates results obtained on the Webvision dataset with the SOP+ Re-labeling method. The first column lists various coreset selection fractions.

Table 6: Performance comparison of various coreset selection methods on WebVision dataset with SOP+ Re-labeling method at various selection fractions of the original dataset. The best results are shown in bold.

|  | k-CenterGreedy | GraNd | Forgetting | SmallLoss | Uniform | Prune4Rel | Moderate | Ours |
|---|---|---|---|---|---|---|---|---|
| 0.05 | 39.46 ± 0.58 | 16.82 ± 0.13 | 40.33 ± 1.35 | 23.44 ± 1.12 | 38.58 ± 0.21 | 43.41 ± 0.33 | 34.51 ± 0.37 | **46.81 ± 1.21** |
| 0.10 | 50.22 ± 1.02 | 22.24 ± 0.56 | 50.41 ± 0.64 | 33.64 ± 1.88 | 47.16 ± 1.88 | 54.82 ± 1.06 | 46.16 ± 1.04 | **56.10 ± 0.06** |
| 0.20 | 59.86 ± 1.74 | 33.32 ± 0.44 | 61.29 ± 1.13 | 46.76 ± 0.04 | 56.38 ± 1.22 | 62.32 ± 1.92 | 55.96 ± 0.88 | **63.32 ± 0.48** |
| 0.30 | 64.64 ± 1.92 | 43.88 ± 1.98 | 65.42 ± 1.14 | 56.94 ± 0.71 | 60.64 ± 1.76 | **67.12 ± 1.52** | 61.48 ± 1.76 | 66.84 ± 0.04 |
| 0.40 | 67.26 ± 0.26 | 52.01 ± 0.24 | 68.92 ± 1.24 | 62.96 ± 1.88 | 64.12 ± 0.18 | 69.38 ± 0.62 | 65.24 ± 0.23 | **69.51 ± 0.18** |
| 0.50 | 69.01 ± 1.36 | 59.28 ± 0.44 | 69.86 ± 0.74 | 65.78 ± 0.15 | 66.92 ± 1.24 | 70.61 ± 0.14 | 66.76 ± 0.18 | **70.77 ± 0.47** |
| 0.60 | 70.76 ± 1.24 | 64.51 ± 1.31 | 70.38 ± 1.02 | 68.92 ± 1.08 | 68.82 ± 1.42 | 71.11 ± 0.86 | 69.52 ± 1.88 | **71.21 ± 0.41** |

## 4.4 Results on ImageNet-1K

Tables 7 and 8 tabulate results obtained on the ImageNet-1K dataset with the SOP+ Re-labeling method with ResNet-50 and ViT architectures, respectively. Our proposed method is able to outperform Prune4Rel at all the coreset selection fractions considered for both the architectures.

## 4.5 Accuracy vs training speed trade-off

Figure 4 depicts the trade-off between accuracy vs. training speed due to coreset selection. It can be seen that coreset selection can improve training speed without significantly impacting full-data set training accuracy.

Table 7: Performance comparison of various coreset selection methods on the ImageNet dataset with SOP+ Re-labeling method at various selection fractions of the original dataset. The best results are shown in bold. ResNet-50 architecture is used. Our proposed method outperforms Prune4Rel across all the coreset selection fractions considered.

| Method | 0.01 | 0.05 | 0.10 | 0.20 | 0.30 | 0.40 |
|---|---|---|---|---|---|---|
| Prune4Rel | $1.71 \pm 0.04$ | $29.20 \pm 0.13$ | $42.22 \pm 0.34$ | $52.03 \pm 0.27$ | $57.18 \pm 0.83$ | $59.09 \pm 0.14$ |
| Ours | $\mathbf{2.33 \pm 0.23}$ | $\mathbf{32.97 \pm 0.29}$ | $\mathbf{47.31 \pm 0.47}$ | $\mathbf{55.42 \pm 0.45}$ | $\mathbf{57.77 \pm 0.75}$ | $\mathbf{60.17 \pm 0.16}$ |

Table 8: Performance comparison of various coreset selection methods on the ImageNet dataset with SOP+ Re-labeling method at various selection fractions of the original dataset. The best results are shown in bold. ViT architecture is used. Our proposed method outperforms Prune4Rel across all the coreset selection fractions considered.

| Method | 0.001 | 0.005 | 0.01 | 0.05 | 0.10 | 0.20 |
|---|---|---|---|---|---|---|
| Prune4Rel | $0.10 \pm 0.07$ | $2.08 \pm 0.05$ | $3.58 \pm 0.11$ | $11.87 \pm 0.05$ | $17.12 \pm 0.23$ | $27.67 \pm 0.31$ |
| Ours | $\mathbf{0.76 \pm 0.03}$ | $\mathbf{2.78 \pm 0.09}$ | $\mathbf{4.79 \pm 0.19}$ | $\mathbf{14.43 \pm 0.08}$ | $\mathbf{22.16 \pm 0.03}$ | $\mathbf{33.93 \pm 0.21}$ |

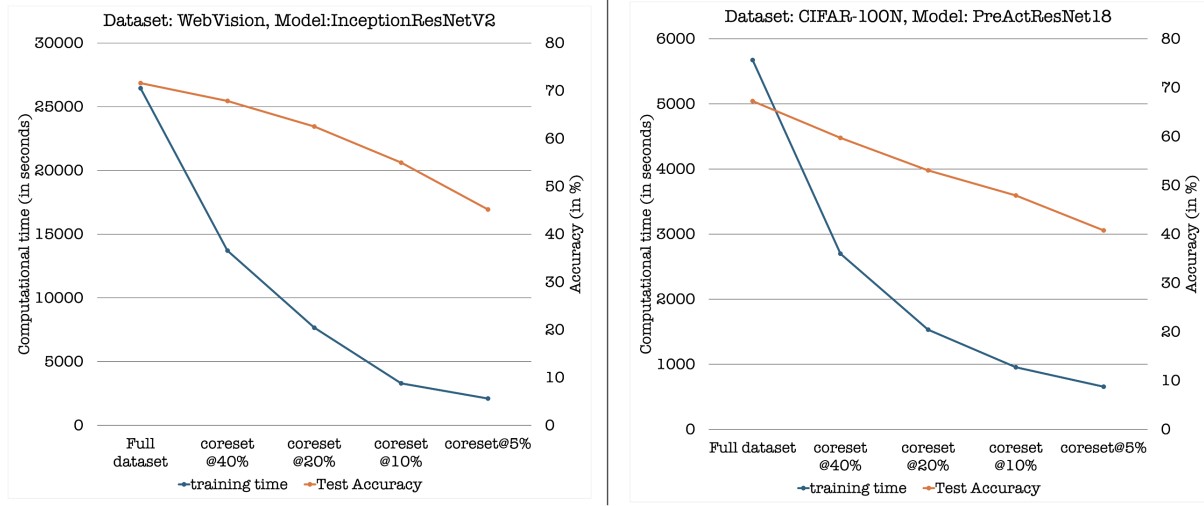

Figure 4: Accuracy vs training speed trade-off at various coreset selection percentages. Training time reduction is much higher than the reduction in accuracy at a lower level of coreset selection percentage, highlighting the benefit of using coreset for re-labeling.

### 4.6 Ratio(%) of noisy examples in the selected subset for CIFAR-100N dataset.

Table 9 compares the ratio of noisy examples in the selected subset for the CIFAR-100N data set. As can be seen, our method progressively selects a higher percentage of noisy examples as the fraction of the subset to be selected increases. Our method progressively but carefully increases the fraction of incorrectly labeled samples in the coreset, which have a higher probability of getting corrected through re-labeling process. This is not the case with methods other than Prune4Rel. SmallL selects a very low percentage of incorrectly labeled samples and therefore sacrifices generalizability. Other methods select a higher percentage of incorrectly labeled samples, and that too, results in poor generalization.

Table 9: Ratio of noisy samples selected as part of the coreset. The percentage of noisy samples selected grows steadily with the increase in the coreset selection percentage.

| Method | 0.2 | 0.4 | 0.6 | 0.8 |
|---|---|---|---|---|
| SmallL | 3.5 | 8.7 | 16.3 | 27.4 |
| Margin | 61.5 | 56.8 | 51.6 | 46.2 |
| Center | 37.5 | 38.7 | 39.9 | 40.4 |
| Forgetting | 37.9 | 34.6 | 33.0 | 36.8 |
| GraNd | 93.9 | 57.3 | 61.2 | 49.3 |
| Moderate | 33.2 | 54.6 | 60.2 | 64.6 |
| Prune4Rel | 28.3 | 29.1 | 33.3 | 37.2 |
| Ours | 13.0 | 18.2 | 23.5 | 29.1 |

### 4.7 Impact on class-wise accuracy

We visualise the class-wise accuracy of training a PreActResNet-18 model on the CIFAR-100N dataset with re-labeling, comparing the performance of the entire dataset and the selection of the coreset with 20% of the dataset in Figure 5. We observe a Spearman rank-order correlation coefficient of 0.802, indicating a high correlation between class-wise accuracies obtained with the full dataset and 20% of the full dataset as a coreset. It can be seen that our method does not adversely impact any particular class during coreset selection. The coreset composed by our method results in class-wise accuracies strongly correlated to those of the full dataset.

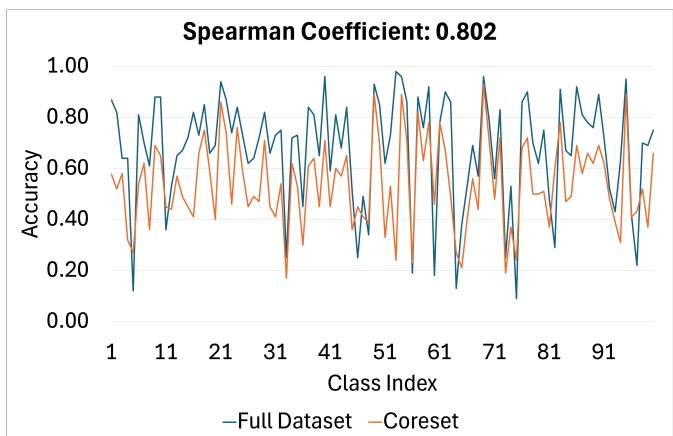

Figure 5: Class-wise accuracy comparison with re-labeling between the full dataset and the coreset with 20 % selection percentage. A high correlation is observed, indicating that the coreset composed by our proposed method does not impact any particular class adversely.

### 4.8 Visualisation of top and bottom-ranked images

Figure 6 shows the top and bottom-ranked images selected in the coreset. As can be seen, the top-ranked images are unambiguous representatives of the 'ostrich' class. In contrast, the bottom-ranked images consist of mislabeled examples and images that do not represent the class.

### 4.9 Comparative analysis with Vision Language Models(VLMs)

To understand the impact of leveraging large multimodal models for correcting noisy annotations, we conducted additional experiments on the ImageNet-1K dataset using a pre-trained vision-language model (VLM,

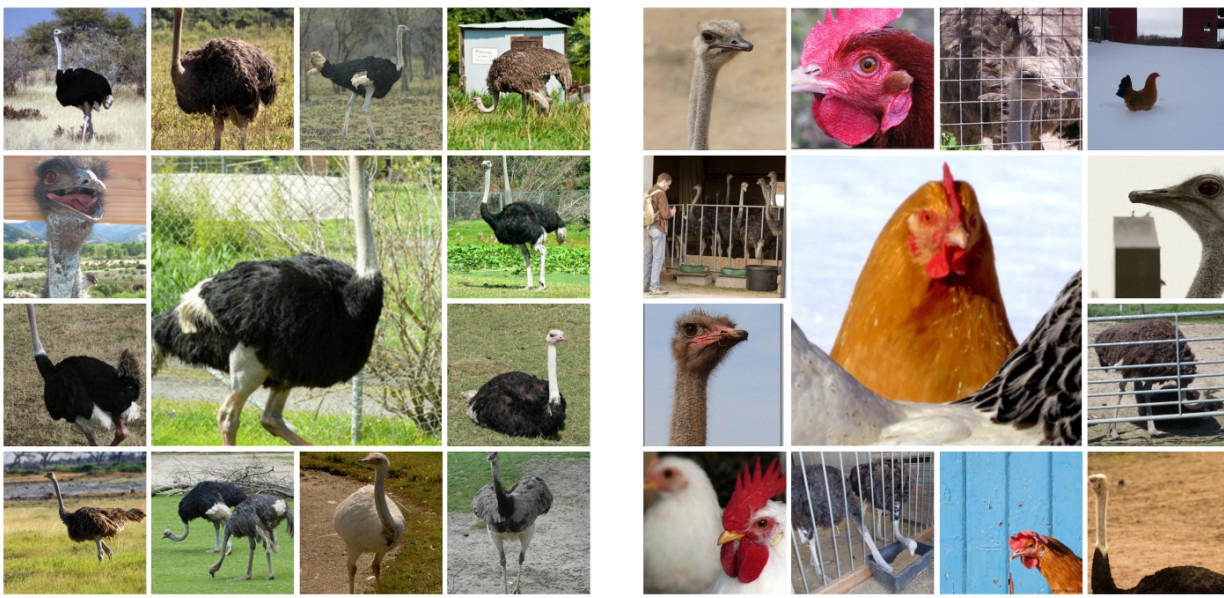

Top ranked images           Bottom ranked images

Figure 6: Top and bottom-ranked images in coreset belonging to class 'Ostrich'. Top-ranked samples are unambiguously representative of their class, while bottom-ranked samples are ambiguous or erroneously labelled.

ViT-H/14) (Ilharco et al., 2021). The VLM was employed as a relabeling mechanism, and its performance was compared with SOP+.

Table 10: Comparative analysis of re-labelling through the VLM model and SOP+. Except for a lower coreset selection fraction of 0.01, SOP+ outperforms the VLM.

| Coreset Fraction | Accuracy with SOP+ | Accuracy with VLM |
|:---:|:---:|:---:|
| 0.01 | $2.33 \pm 0.23$ | $\mathbf{6.87 \pm 0.14}$ |
| 0.05 | $\mathbf{32.97 \pm 0.29}$ | $20.21 \pm 0.23$ |
| 0.10 | $\mathbf{47.31 \pm 0.47}$ | $27.79 \pm 0.92$ |
| 0.20 | $\mathbf{55.42 \pm 0.45}$ | $37.60 \pm 0.51$ |

From the results provided in Table 10, we observe that while the VLM achieves higher accuracy than SOP+ at very low coreset fractions (0.01), SOP+ consistently outperforms the VLM for larger fractions (0.05–0.20). This indicates that although large multimodal models provide some advantage when very limited data is available, specialized re-labeling methods such as SOP+ remain more effective and robust as the available data fraction increases. These findings reinforce our design choice of treating re-labeling as a black-box and focusing on the coreset contribution, while also highlighting that future work could explore hybrid strategies that combine coreset-driven relabeling with the representational strength of VLMs.

## 4.10 Ablation studies

### 4.10.1 Selection threshold

The selection threshold on the gradient similarity for neighborhood identification, $\Phi$, is important in generalization performance. Figure 7 compares the accuracy obtained at coreset selection of 5% and 10% obtained at various selection threshold values $\Phi$. We use a threshold of 0.4 in our experiments.

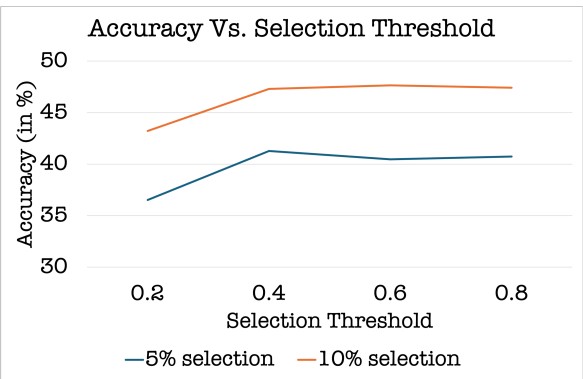

Figure 7: Ablation study of accuracy vs. selection threshold. Results are obtained on the CIFAR-100N dataset with the PreAct ResNet-18 model. Accuracy varies with selection threshold $\Phi$, and best results are achieved in the $[0.4, 0.6]$ range.

### 4.10.2   Confidence score

We have tweaked the vanilla implementation of 'Noise-free gradients' by introducing a confidence score as a weight during the selection of the coreset. Figure 8 compares the accuracy obtained at various coreset selection percentages for the vanilla implementation and with confidence score as weights. A clear improvement can be noticed by incorporating the confidence score in selecting the coreset.

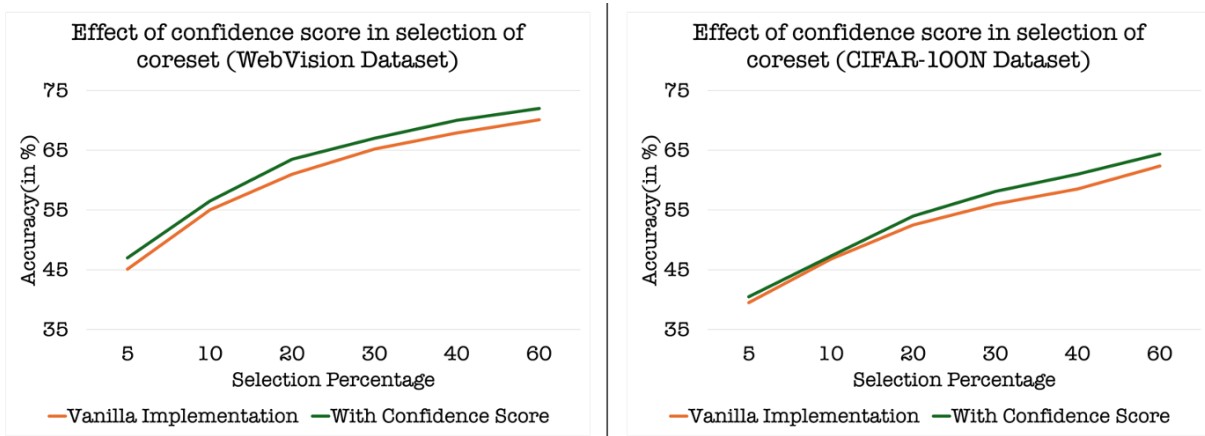

Figure 8: Impact of confidence score on accuracy. A clear improvement can be noticed by incorporating the confidence score when selecting the coreset. The left image shows results for the WebVision dataset with the InceptionResNetV2 architecture. The right image shows results for the CIFAR-100N dataset with the PreAct ResNet-18 architecture.

### 4.10.3   Variation of softmax

Softmax outputs tend to be overconfident, even when the model's predictions are incorrect (Guo et al., 2017; Nguyen et al., 2015). Our application of softmax is geared towards understanding the relative importance of sampling. Our choice of using softmax as a confidence score is motivated by its simplicity and empirical effectiveness in capturing mislabelling patterns in visually similar classes (Figure 3). We refine the selection of the coreset by multiplying the number of neighbors by the confidence score of the samples.

To understand whether the variation in softmax distribution impacts generalization accuracy significantly, we have carried out temperature scaling in the softmax calculation (Guo et al., 2017) for comparative analysis.

The Table 11 compares accuracy values obtained on the CIFAR100 dataset with the SOP+ re-labelling method for various coreset selection percentages and temperature values.

Table 11: Impact of softmax temperature on accuracy for various selections of coreset fraction for the CIFAR100 dataset with SOP+ re-labelling method. Over a wide range of temperature values, classification performance is not affected, underscoring the fact that choice of softmax as a confidence score is not detrimental in our proposed method.

| Coreset Fraction | T=0.1 | T=1.0 | T=10.0 | T=30.0 |
|:---:|:---:|:---:|:---:|:---:|
| 0.05 | $40.90 \pm 0.30$ | $40.48 \pm 0.02$ | $40.17 \pm 0.08$ | $39.63 \pm 0.24$ |
| 0.10 | $46.74 \pm 0.23$ | $46.82 \pm 0.34$ | $46.83 \pm 0.03$ | $46.30 \pm 0.01$ |
| 0.20 | $53.13 \pm 0.09$ | $53.92 \pm 0.24$ | $53.10 \pm 0.10$ | $51.98 \pm 0.04$ |
| 0.30 | $56.94 \pm 0.13$ | $58.04 \pm 0.25$ | $56.50 \pm 0.10$ | $57.10 \pm 0.30$ |
| 0.40 | $59.73 \pm 0.30$ | $60.55 \pm 0.17$ | $59.45 \pm 0.34$ | $59.40 \pm 0.40$ |

With a wide range of temperature values, classification performance is not affected. Hence, it shows that the proposed softmax approach is effective in the current setting.

## 5 Conclusion

In this work, we presented an adaptation of an existing coreset selection method tailored to improve learning from noisy datasets. In this approach, we select a coreset composed of samples with the highest number of neighbours with high gradient similarity (measured by cosine similarity), weighted through their confidence score. Experimental results demonstrate that our method effectively mitigates the impact of label noise, leading to improved performance compared to standard coreset approaches. Particularly, performance gains are significant at lower coreset sizes, because at a very small coreset size, the purity of labels and reliability of the samples strongly impact classifier accuracy. Our findings open avenues for further research into confidence-aware sampling strategies in noisy learning environments.

Hyper-parameter tuning and other tasks that require multiple training runs over the entire dataset are time-intensive. The enhancement achieved in execution time by our method makes it a preferred approach for these tasks. The diversity of the selected coreset is an area of interest for future research. We wish to explore techniques to improve diversity and its impact on the generalization performance of the chosen coreset. Exploring hybrid strategies that combine coreset-driven relabeling with the representational strength of VLMs, is also an interesting direction of future research.

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
