# OpenReview forum: "Coreset-Driven Re-Labeling: Tackling Noisy Annotations with Noise-Free Gradients"
_TMLR — Accepted by TMLR_

### Review · Reviewer_2RBt · 2025-07-07

**Summary Of Contributions:**

This paper proposes an improved coreset selection method aimed at enhancing the efficiency and effectiveness of re-labeling techniques for learning from noisy labeled datasets. Specifically, the authors build on the existing “Noise-free Loss Gradients” method by introducing a confidence score based on the model’s softmax outputs to weight gradient similarity and select a more representative subset for training. The proposed approach is evaluated on CIFAR-100N, WebVision, and ImageNet-1K, combined with two popular re-labeling methods (DivideMix and SOP+). The experimental results demonstrate that the method can improve classification accuracy while significantly reducing computational cost.

**Audience:**

Yes

**Claims And Evidence:**

Yes

**Requested Changes:**

- Explore or at least discuss the use of more robust uncertainty measures to mitigate the limitations of softmax confidence under label noise.
- Provide theoretical justification or empirical heuristics for choosing Φ to improve reproducibility and robustness.
- Consider analyzing or experimenting with other tasks beyond image classification to enhance the generality of the method.
- Introduce mechanisms that encourage diversity in the selected coreset, particularly for imbalanced datasets.

- How does your method fundamentally differ from Prune4ReL in terms of mechanism, confidence definition, and application scenario?
- Why did you choose to use softmax confidence directly instead of more robust uncertainty estimation methods (e.g., entropy, MC Dropout, deep ensembles)?
- Can you provide any theoretical insights or automated procedures to guide the selection of the threshold Φ?
- Have you considered applying your method to tasks beyond image classification? Do you have any preliminary results or theoretical arguments supporting its generalizability?
- Have you considered incorporating sample diversity constraints into the coreset selection to improve performance on class-imbalanced datasets?
- Were all the baseline results in the comparative experiments reproduced by the authors? If so, could the authors clarify how the baseline results in Table 4 of this paper align almost exactly (with discrepancies smaller than 0.1) with the corresponding baseline results reported in Table 1 of the Prune4ReL paper? This raises questions about whether these results were independently reproduced or directly referenced, and clarification would improve the transparency and credibility of the experimental setup.

**Strengths And Weaknesses:**

Strengths
- The paper addresses an important practical challenge of improving computational efficiency when training on noisy-labeled datasets.
- The integration of confidence scores into gradient similarity is straightforward and easy to implement.
- The method is thoroughly tested on multiple benchmark datasets with different re-labeling methods and various coreset baselines.
- The method delivers noticeable training speedups at low coreset selection fractions, with manageable drops in accuracy.
- The paper includes ablation studies, class-wise accuracy analysis, and sample visualizations to enhance result interpretability.

Weaknesses
- Limited Novelty (Key Issue): This work is not the first to combine coreset selection with re-labeling. The prior work Prune4ReL (Park et al., 2023) has already introduced and systematically studied this problem. The current paper represents an incremental improvement over this existing approach rather than a fundamental innovation.
- Simplistic Confidence Definition: It is well established that softmax outputs tend to be overconfident, even when the model's predictions are incorrect (Guo et al., 2017; Nguyen et al., 2015). This limitation is particularly problematic in noisy label scenarios. The paper does not acknowledge or address this known issue, which may undermine the reliability of the proposed confidence-based selection.
- Lack of Theoretical Support: The method lacks theoretical analysis, particularly regarding the choice of the key hyperparameter Φ (the gradient similarity threshold), which is selected empirically without justification.
- Narrow Task Scope: The method is only evaluated on image classification tasks. Its applicability to other domains or tasks (e.g., object detection, NLP) is not addressed.
- Lack of Diversity Consideration: The coreset selection focuses solely on representativeness but does not encourage sample diversity, which may negatively affect performance on imbalanced datasets.

References

[1] Guo C, Pleiss G, Sun Y, et al. On calibration of modern neural networks[C]//International conference on machine learning. PMLR, 2017: 1321-1330.

[2] Nguyen A, Yosinski J, Clune J. Deep neural networks are easily fooled: High confidence predictions for unrecognizable images[C]//Proceedings of the IEEE conference on computer vision and pattern recognition. 2015: 427-436.

---

> ### Author Response · Authors · 2025-07-18
> **Rebuttal to Reviewer 2RBt**
>
> We thank the reviewer for their thorough and insightful feedback. Below, we address each point raised in the review, including weaknesses and requested changes.
>
>
> **W1**: We agree that this work is not the first to combine coreset selection with re-labeling. We don't claim so either. However, the proposed solution is different from that of Prune4Rel for the same problem. Our work adapts gradient similarity-based coreset selection, integrating a confidence score to enhance coreset selection by prioritizing samples likely to be mislabeled due to visual similarity. Prune4Rel utilizes neighborhood confidence maximization. Our method leverages model confidence to focus on samples with high re-labeling potential. We have shown non-trivial improvement over Prune4Rel and other coreset selection methods through extensive experimentation.
>
>
> We believe our work is of interest to the TMLR audience. We have also provided anonymized code for reproduction.
>
> **W2**. We thank the reviewer for their insight into the limitations of softmax outputs. We acknowledge that softmax tends to be overconfident, but our application of softmax is geared towards understanding the relative importance of sampling. Our choice of using softmax as a confidence score is motivated by its simplicity and empirical effectiveness in capturing mislabelling patterns in visually similar classes (Figure 3). We refine the selection of the coreset by multiplying the number of neighbors by the confidence score of the samples.
>
> We have carried out temperature scaling in the softmax calculation (Guo et al., 2017) for comparative analysis. The table below provides a comparison of accuracy values obtained on the CIFAR100 dataset with the SOP+ re-labelling method for various coreset selection percentages and temperature values.
>
> | Coreset Fraction | T = 0.1    | T = 1.0    | T = 10.0   | T= 30.0   |
> | ---------------- | ------------- | ------------- | ------------- | ------------- |
> | 0.05             | 40.90 ±0.30 | 40.48 ± 0.02 | 40.17 ± 0.08 | 39.63 ± 0.24 |
> | 0.10             | 46.74 ±0.23 | 46.82 ± 0.34 | 46.83 ± 0.03 | 46.30 ± 0.01 |
> | 0.20             | 53.13 ±0.09 | 53.92 ± 0.24 | 53.10 ± 0.10 | 51.98 ± 0.04 |
> | 0.30             | 56.94 ±0.13 | 58.04 ± 0.25 | 56.50 ± 0.10 | 57.10 ± 0.30 |
> | 0.40             | 59.73 ±0.30 | 60.55 ± 0.17 | 59.45 ± 0.34 | 59.40 ± 0.40 |
>
> As can be observed from the above table, with various range of temperature values, classification performance is not affected. Hence, it shows that the proposed approach of softmax is effective in the current setting.
>
> **W3**. The gradient similarity threshold is a hyperparameter (which is not uncommon in ML), obtained heuristically through extensive testing. Figure 7 of the draft compares accuracy vs. various selection thresholds.
>
> **W4**. Our current work is focused on image classification. Due to the prevalence of noisy labels in large-scale image datasets (e.g., CIFAR100N, Web Vision, etc.) and established re-labelling benchmarks (DivideMix and SOP+), we have focused on image classification applications. There have been various works on the utilization of coreset for other domains, and these are separate, stand-alone works in their own right. We will explore the utilization of our proposed method for other domains such as NLP and object detection.
>
>
> **W5**. Lack of diversity consideration: As we have mentioned in Section 5 (Conclusion), diversity of the selected coreset is an area of interest for future research.

---

> > ### Author Response · Authors · 2025-07-18
> > **Rebuttal to Reviewer 2RBt (Contd.)**
> >
> > **Addressing requested changes**
> >
> > **C1**.  We have provided the impact of temperature on softmax-based confidence score calculation on the accuracy of the test dataset of CIFAR100N with SOP+ re-labelling method.
> >
> > **C2**. The gradient similarity threshold is a hyperparameter obtained heuristically through extensive testing. Figure 7 of the draft compares accuracy vs. various selection thresholds.
> >
> >
> > **C3**. We will consider other tasks in our future work, as these are standalone works in their own right.
> >
> > **C4**. We will incorporate a subsection in section 3 of the draft to discuss the difference between Prune4Rel and our proposed methodology. These are reproduced here for easier reference.
> >
> > **Mechanism**: Prune4Rel maximizes total neighborhood confidence to select coresets, focusing on local label consistency. Our method adapts the noise-free gradient framework for coreset selection, weighted by per-sample confidence scores.
> >
> > **Confidence definition**: Prune4Rel uses neighborhood-based confidence, while we use softmax confidence (i.e., model confidence on each sample).
> >
> > **C5**. Our application of softmax is geared towards understanding the relative importance of sampling. Our choice of using softmax as a confidence score is motivated by its simplicity and empirical effectiveness in capturing mislabelling patterns in visually similar classes (Figure 3). We augment the selection of the coreset by multiplying the number of neighbors by the confidence score of the samples.
> >
> > **C6**. As shown in Figure 7, the threshold value in the range of [0.4,0.6] provides the best accuracy for various selection percentages.
> >
> > **C7**. We will consider the application of our approach to other tasks in our future work.
> >
> > **C8**. As mentioned in section 5, we will consider sample diversity constraints in coreset selection in our future work.
> >
> > **C9**. Baseline results were independently reproduced using the code provided by the authors.

---

### Review · Reviewer_fLBQ · 2025-07-13

**Summary Of Contributions:**

The paper studies the problem of re-labeling for noise-label classification. Particularly, the paper proposes a coreset-based method, which first identifies a small coreset of crucial samples and then performs relabeling correction only on the small coreset. To construct the coreset, the paper builds upon a previous approach, which first identifies samples that are most similar to their neighbors and then weights them by their prediction confidence. Empirically, the proposed method can significantly outperform baseline coreset selection, especially for small coreset sizes.

**Audience:**

Yes

**Claims And Evidence:**

Yes

**Requested Changes:**

Please address the weakness part.

**Strengths And Weaknesses:**

Strengths:
1. The proposed method is simple and easy to implement. The method shows significantly improved performance when the coreset size is small.

Weaknesses:
1. The proposed method is a marginal modification of the previous method on coreset selection. Moreover, the intuition behind the modification is not very clear. The paper discusses that mislabeled samples are likely to be predicted with high confidence. However, the empirical evidence for that claim is insufficient. Ideally, we should observe some correlation between the prediction confidence and the mislabeled rate to make the claim more solid.

2. More baselines could be added in the experiments. E.g., comparisons with submodular-based methods such as CRAIG or simply using submodular functions to construct the coreset. Also, it is useful to provide comparisons with other non-coreset noise-label learning methods as a reference.

3. What if the noise has a particular systematic bias? For example, in the extreme case, all the representative samples are labeled correctly, and only the samples that are close to the boundaries are mislabeled. In such a case, as the method focuses on samples that are similar to most of their neighbors and are therefore representative, it may be the case that very few noisy labels are caught.

---

> ### Author Response · Authors · 2025-07-18
> **Rebuttal to Reviewer fLBQ**
>
> We thank the reviewer for their thoughtful comments and constructive suggestions.
>
> **W1**. We acknowledge that our method builds upon the noise-free gradient framework by Mohanty et al.(2025). Our Primary contribution lies in adapting a representation-focused coreset technique to the noisy label regime, where confidence can serve as a proxy for correct likelihood. The modified similarity score prioritizes not just neighborhood agreement, but the model’s belief in those neighbors, crucial in noisy settings. Our method prioritizes samples with high re-labeling potential due to visual similarity between classes.
>
> The paper discusses that mislabeled samples that are visually similar to the correct class will be given a high confidence score by a trained model. We would like to bring the attention of the reviewer to Figure 3.
>
> W2. We have conducted experimentation on the Craig method, and the table below compares the performance of Craig vs our proposed method. The results are given for the CIFAR100N dataset with the SOP+ method.
>
> | Fraction | Craig  | Our Method |
> | ------------ | ------------- | ----------------- |
> | 0.05               | 23.25 (±2.61) | **40.48 (±0.02)**    |
> | 0.10               | 26.22 (±1.56) | **46.82 (±0.34)**    |
> | 0.20               | 45.34 (±3.50) | **53.92 (±0.24)**     |
> | 0.30               | 51.70 (±2.88) | **58.04 (±0.25)**     |
> | 0.40               | 55.23 (±2.14) | **60.55 (±0.17)**    |
> | 0.50               | 58.07 (±2.24) | **62.76 (±0.17)**     |
> | 0.60               | 60.04 (±1.23) | **64.38 (±0.05)**     |
>
> Our method is consistently outperforming CRAIG, as can be seen from the above table, achieving significant performance improvement in the lower selection fractions.
>
> **W3**. We thank the reviewer for this insightful point. We acknowledge that systematic bias in noise distribution could affect coreset selection. We would like to emphasize that we have focused on real-world datasets where the noisy labels are caused by human annotation errors (CIFAR-100N and WebVision) and one large and complex dataset like Imagenet-1K, with asymmetric noise where noise is added randomly to the classes which are visually similar (classes which are neighbors in WordNet hierarchy). As Table 8 shows, our method progressively increases the percentage of noise samples to better aid the re-labelling methods, and hence boundary-focused noise patterns may be a potential limitation.

---

### Review · Reviewer_mvgh · 2025-07-14

**Summary Of Contributions:**

This paper proposes a improved version of gradient similarity based coreset selection method to deal with label noise. The idea is first to select the mislabeled but high similar concepts into the corset (as they improve generalization) and then use a sample-wise weighted confidence score to find the most suitable coreset samples. Experiments on typical datasets such as CIFAR-100N, ImageNet-1k, WebVision were conducted to verify the effectiveness of the proposed approach.

**Audience:**

Yes

**Claims And Evidence:**

No

**Requested Changes:**

1. Evaluate the proposed method within the large model fine-tuning paradigm.

2. Assess performance using state-of-the-art models on the SOTA benchmark leaderboards.

3. Test the method on larger-scale and more contemporary datasets, as ImageNet-1k is now somewhat outdated.

4. Compare against VLM-based annotation correction approaches, such as querying an open-source VLM for class labels and confidence scores.

**Strengths And Weaknesses:**

Strengths:

1. The paper addresses an important challenge in large-scale real-world datasets: annotation noise.

2. The proposed method is notably simple, offering advantages in terms of ease of implementation.

Weaknesses:

1. The problem formulation and experimental setup are outdated in the era of large models. The authors claim that label noise is a critical issue in large-scale datasets, yet none of the tested datasets are sufficiently large to substantiate this claim. Evaluations on truly large-scale datasets are necessary to demonstrate real-world relevance.

2. The practical significance is limited. Most contemporary large-scale datasets, such as LAION, no longer rely on labels but instead use text descriptions and are largely unsupervised. It is unclear how the proposed method would address noise in such settings.

3. The experimental setup and learning paradigm lack real-world alignment. The authors argue that label noise impacts supervised learning, but the best ImageNet performances now come from fine-tuning large pre-trained models, not training from scratch. Fine-tuning is also more appropriate for coreset learning.

4. The models used in the experiments are outdated. The authors should evaluate their method using state-of-the-art model architectures and benchmarks.

5. The paper does not explore the possibility of leveraging large multimodal models to correct noisy annotations.

6. Performance improvements are marginal under several noise ratios.

7. Technical novelty is limited; the method is essentially a weighted adaptation of an existing approach.

8. Most cited works are from 2022 or 2023, failing to account for recent paradigm shifts brought by large models. The conventional setup adopted has become less meaningful nowadays.

9. Coreset selection is somewhat similar to the few high-quality examples used in few-shot learning or supervised fine-tuning of a pre-trained large foundation model. I would suggest this setting for this paper.

---

> ### Author Response · Authors · 2025-07-22
> **Rebuttal to reviewer mvgh**
>
> We thank the reviewer for their insightful and constructive feedback. Below we address the weaknesses and requested changes raised by the reviewer.
>
> W1. Our primary goal is to study coreset selection for re-labelling application in image classification. We have considered two real-world noisy label datasets. We have also experimented on ImageNet-1K dataset, which is a benchmark dataset for image classification applications. We believe that ImageNet-1K dataset, with approximately 1.3 million images (and at a 20% noise rate, 256k images with noisy labels), is sufficiently large-scale to validate our approach.
>
> W2. Our work is intended for image classification applications, and the re-labelling methods that we have utilized are SOTA for dealing with label noise.
>
> W3. There are multiple settings where supervised learning finds its place. In fine tuning VLMs, noisy label setting is still valid.
>
> W4. While we do agree that ResNet-50 and similar models are not state-of-the-art, they are not irrelevant or outdated, as they find their usage in memory-constrained settings. As one of the major goals of coreset selection is to deal with model training in a memory-constrained setup, we believe the settings we have used demonstrate the effectiveness of our proposed methodology.
> We will present the results with transformer architecture like ViT in the camera-ready submission.
>
> W5. As we have brought out in Figure 1, we utilize the Re-labelling method as a black box to study the impact of coreset on such an application. Furthermore, we have implemented correction of noisy annotations on ImageNet-1K dataset, using a pre-trained VLM model (ViT-H14), and accuracy comparison is given in the table below.
>
> | Coreset Fraction | Accuracy with SOP+| Accuracy with VLM  |
> | ---------------- | ------------------------- | ------------------------ |
> | 0.01             | 2.33 ± 0.23              | 6.87 ± 0.14             |
> | 0.05             | 32.97 ± 0.29             | 20.21 ± 0.23            |
> | 0.10             | 47.31 ± 0.47             | 27.79 ± 0.92            |
> | 0.20             | 55.42 ± 0.45             | 37.60 ± 0.51            |
>
> As can be seen above, SOP+ outperforms the VLM model for all the fractions except 0.01.
>
> W6. Across various coreset selection fractions, our method achieves an average increase in accuracy of 4.22% on the CIFAR-100N dataset with the DivideMix re-labelling method.
>
> W7. Respectfully, we like to mention that “novelty” of a method is not a necessary criterion for acceptance for TMLR. We believe our claims in the paper are supported by accurate and clear evidence, and our work will be of interest to TMLR’s audience.
>
> W8. We respectfully disagree with the reviewer that conventional setup has become less meaningful. Many real-world applications still rely on noisy datasets, and as we have shown in the above table, the conventional setup with SOP+ re-labelling method outperforms VLM-based noisy label correction. While we acknowledge evolution of learning paradigms in the era of foundation models, we believe our proposed setup and evaluation protocol remain meaningful and effective, especially in resource-constrained contexts.

---

### Decision · Action_Editor_9JJ6 · 2025-09-04

**Recommendation:** Accept with minor revision

**Additional Comments:**

The rebuttal addressed most of the reviewers concerns. However, please update the final version of the paper with the responses to the reviewers.

**Audience:**

Yes

**Audience Explanation:**

The AE believes that some individuals in the TMLR audience would be interested in this paper, as it demonstrates a simple and effective way to improve image classification under the noisy-label scenario. This is an application which is still common in practice.

However, the AE does also agree with the reviewers' comments that this paper is not as applicable today, with the prevalence of LLMs and VLMs. Such models are not trained for image classification, and do not use class labels in their training data, but rather language in the form of captions or (instruction, answer) pairs. This paper would have a lot more impact on the community if it was applicable in these scenarios, and as Reviewer mvgh stated, if the authors could connect their work to Supervised Finetuning (SFT) or few-shot prompting which typically rely on a few high-quality examples.

Nevertheless, on the balance, the AE does think that this paper would be of interest to some individuals in the TMLR audience.

**Claims And Evidence:**

Yes

**Claims Explanation:**

This paper addresses the problem of image classification with noisy training labels, using a coreset selection method. Concretely, the authors build on the existing "Noise-free Loss Gradients" method, and add a confidence score based on the model's softmax outputs. This  confidence score is used to weight the gradient similarity to choose a more representative coreset for training.

Reviewers appreciated the fact that the method is simple and easy to implement, runs quickly, and achieves strong results on a number of noisy image classification datasets such as ImageNet, Webvision and Noisy CIFAR-100.

Whilst the authors claims are supported well experimentally, reviewers had some concerns regarding the novelty of the approach, as it is quite similar to the existing "Noise-free Loss Gradients" work, and there is not much underlying theoretical justification for the method either.